# Impact of green clay authigenesis on element sequestration in marine settings

Andre Baldermann [1✉], Santanu Banerjee [2], György Czuppon[3], Martin Dietzel [1], Juraj Farkaš[4], Stefan Löhr [5], Ulrike Moser[1], Esther Scheiblhofer[1], Nicky M. Wright [6] & Thomas Zack [4,7]

Retrograde clay mineral reactions (reverse weathering), including glauconite formation, are first-order controls on element sequestration in marine sediments. Here, we report substantial element sequestration by glauconite formation in shallow marine settings from the Triassic to the Holocene, averaging $3 \pm 2$ mmol·cm$^{-2}$·kyr$^{-1}$ for K, Mg and Al, $16 \pm 9$ mmol·cm$^{-2}$·kyr$^{-1}$ for Si and $6 \pm 3$ mmol·cm$^{-2}$·kyr$^{-1}$ for Fe, which is ~2 orders of magnitude higher than estimates for deep-sea settings. Upscaling of glauconite abundances in shallow-water (0–200 m) environments predicts a present-day global uptake of ~$\leq 0.1$ Tmol·yr$^{-1}$ of K, Mg and Al, and ~0.1–0.4 Tmol·yr$^{-1}$ of Fe and Si, which is ~half of the estimated Mesozoic elemental flux. Clay mineral authigenesis had a large impact on the global marine element cycles throughout Earth's history, in particular during 'greenhouse' periods with sea level highstand, and is key for better understanding past and present geochemical cycling in marine sediments.

[1] Institute of Applied Geosciences, Graz University of Technology, NAWI Graz Geocenter, Graz, Austria. [2] Department of Earth Sciences, Indian Institute of Technology Bombay, Powai, Mumbai, India. [3] Institute for Geological and Geochemical Research, Research Centre for Astronomy and Earth Sciences, Eötvös Loránd Research Network, Budapest, Hungary. [4] Department of Earth Sciences, Metal Isotope Group (MIG), University of Adelaide, North Terrace, Adelaide, SA, Australia. [5] Department of Earth and Environmental Sciences, Macquarie University, Sydney, NSW, Australia. [6] Earthbyte Group, School of Geosciences, University of Sydney, Sydney, NSW, Australia. [7] Department of Earth Sciences, University of Gothenburg, Göteborg, Sweden. ✉email: baldermann@tugraz.at

Chemical elements are supplied to the global ocean by the chemical and physical weathering of carbonate and silicate minerals on the continents, and the subsequent transport of dissolved and particulate matter by rivers, groundwater, glaciers, and wind[1–3]. Hydrothermal sites of mid-oceanic ridges and flanks constitute another major source of chemical elements to the oceanic dissolved pool, but hydrothermal reactions between seawater and the oceanic crust may also result in a significant element fixation and burial[4–7]. The dissolution of continent-derived reactive particulate matter and the subsequent uptake of dissolved elements by reverse weathering (i.e., clay mineral authigenesis), occurring in both shallow marine and deep marine settings, are other key factors that control the rate and magnitude of element cycling in marine settings[8,9]. It is thought that the mean elemental fluxes close to the sediment-seawater interface are controlled mainly by the tectonic setting, sediment provenance, and climate regime prevailing on the continents[2,10,11], which determine the sedimentation rate, composition, and reactivity of the continental weathering influx. This influx and the corresponding chemical evolution of the oceans, as well as long-term variations of the atmospheric carbon dioxide ($CO_2$) pool, subsequently influence the marine silicate and carbonate bioproductivity[3,12–17].

The source-sink relations of the global elemental cycles are increasingly well constrained due to advances in e.g., high-precision isotope and element concentration measurements in benthic chambers, novel isotopic tracing methods, and isotope-enabled earth system models combined with multivariate statistical modeling[3,15,18–23]. However, there remain large gaps in knowledge concerning, for example, the marine rare earth elements (REE), trace elements, and potassium (K) budgets[7,9,24,25]. Moreover, the magnitude of element burial attributable to reverse weathering and (green) clay mineral authigenesis on the ocean floor remains poorly constrained, except for very specific settings (e.g., well studied deltaic sediments), and is only indirectly accounted for in earth system models[8,14,21,26].

The long-standing view that clay mineral reactions taking place at low to ambient temperature (≤30 °C) over much of the Earth's surface are very slow has been increasingly challenged[21,26–29]. Fast retrograde clay reactions occur in hydrothermal settings or in deep-burial and diagenetic surroundings[5,7], in deltaic sediments, mangrove forests or estuaries[21,24,27], and even in the deep-sea[3,15,20,28] are increasingly considered critical controls on element sequestration in modern[30] and ancient marine sediments[31,32]. However, the often cryptic nature of authigenic clays, as well as their small particle size and compositional similarity to detrital clays, make it difficult to estimate their abundance in the rock record, so that the broader significance of authigenic clays in the marine geochemical cycle remains disputed.

The mineral glauconite, (K, Na, Ca)(Fe, Al, Mg)[(Si,Al)$_4$O$_{10}$] (OH)$_2$, is one such authigenic clay, which commonly forms large (mm-scale), distinct, and easily recognized green granules near the sediment-water interface, making it a particularly important proxy to assess the broader significance of clay authigenesis in marine sediments. Glauconite forms in siliciclastic and calcareous sediments in marine and continental depositional environments, and within a wide range of substrates, including fecal pellets, foraminifera chambers, and lithoclasts[33–36]. It is thought to evolve from K-poor, but iron (Fe)-rich smectite to K- and Fe-rich glauconite via the formation of glauconite-smectite intermediates over a time-frame less than a few million years (Myr) after sediment deposition[36]. While Fe uptake by glauconite and glauconite-smectite formation in modern deep-sea settings has been shown to be significant, up to six-fold greater than Fe sequestration by pyrite formation in near-surface sediments[28], there has been no attempt to estimate the broader impact of glauconite formation on past and present marine geochemical cycles.

In this study, we fill this gap using a well-characterized, Cretaceous-aged glauconite-bearing sequence from Langenstein, Northern German basin, to calculate elemental sequestration rates related to glauconite formation in shallow-water settings. The Langenstein sequence (Fig. 1) is an authigenic glauconite deposit formed in a palaeo-shelf setting[37–39]. Here, shallow-water carbonate or sandstone lithologies contain abundant glauconite, with overlying shelf sediments hosting smaller quantities of glauconite (see Supplementary Fig. 1 for lithofacies analysis). K-Ar dating indicates glauconite formation was completed within <1 Myr of deposition, close to the sediment-seawater interface[40]. This study site is representative of many modern and palaeo-shelf settings that accumulate glauconite minerals[33,35,41–45]. Upscaling of the Langenstein rates indicates that element sequestration through green clay authigenesis strongly impacted the marine geochemical cycle throughout Earth's history.

## Results and discussion

**Characterization of Langenstein glauconite**. XRD analysis identifies the green grains within the sandstone and carbonate lithologies as glauconite with minor admixtures of glauconite-smectite based on broad reflections at 10 Å (001), 5.0 Å (002), 4.5 Å (020), 3.3 Å (003), 2.6 Å (13$\bar{1}$), and 1.51 Å (060,33$\bar{1}$) (Fig. 2a). Well defined reflections at 3.6 Å (11$\bar{2}$) and 3.1 Å (112), and the weak "XRD hump" between 25 to 40° 2Θ indicate the green grains are mixtures of the 1 M and 1 M$_d$ polytype structures, which correspond to ordered glauconite and disordered glauconite-smectite[46].

Petrographically, the green grains are dominated by glauconitized fecal pellets of dark green and medium green color (~85 ± 5 wt%; insert in Fig. 2b), with subordinate light green colored glauconitized fecal pellets (~10 ± 5 wt%) and greenish infills in foraminifera chambers (~≤5 wt%). Backscattered electron imaging illustrates the micro-texture of the glauconite is made of tightly-packed, sub-micron-sized crystals forming "rosette-like" structures (Fig. 2b). The high-resolution TEM lattice fringe image shows the flake-shaped glauconite particles are made of 10 Å domains (Fig. 2c and the insert). All these features are indicative of evolved, mature glauconite[33,46].

Glauconite commonly evolves from an authigenic Fe-smectite precursor[46], with glauconite maturation from a K-poor but Fe(III) rich nascent stage (<4 wt% $K_2O$) to a K-rich evolved or highly evolved stage (>8 wt% $K_2O$) occurring over less than a few Myr. Chemically, the vast majority (more than 95% of the chemical data, see Supplementary Table 1) of the glauconite grains has $K_2O$ contents exceeding 7 wt%, frequently reaching up to >9 wt% (Fig. 2d), which is characteristic of mature glauconite (i.e., reflecting the "evolved" to "highly evolved" stage, according to the glauconite maturity classification of ref. [33]). The total Fe contents (defined here as TFe; a sum of $Fe_2O_3$ and FeO) range from 16 to 26 wt% (Fig. 2e), which shows the glauconites are Fe-rich[45]. The measured aluminum (Al), magnesium (Mg), and silicon (Si) contents (expressed as oxides) are in the typical range of Mesozoic and Cenozoic glauconites[34,35], averaging 7.9 ± 1.0 wt% $Al_2O_3$, 4.1 ± 0.3 wt% MgO, and 51.8 ± 1.5 wt% $SiO_2$, respectively. The sodium (Na) (<0.2 wt% $Na_2O$) and calcium contents (<0.2 wt% CaO, but up to 7.9 wt% CaO where hydroxyl-apatite and calcite inclusions are present) are generally low, as expected in mature glauconite (Supplementary Table 1). The calculated structural formula of the glauconites varies in the range of $(K_{0.75-0.82}Ca_{0.01-0.04}Na_{0-0.01})$ $(Fe^{3+}_{1.06-1.20}Fe^{2+}_{0.11-0.12}Al_{0.29-0.42}Mg_{0.40-0.46})[Si_{3.65-3.73}Al_{0.27-0.35}$ $O_{10})](OH)_2$ based on averaged chemical data obtained from "pure" glauconite (i.e., inclusion-free glauconite; see Supplementary

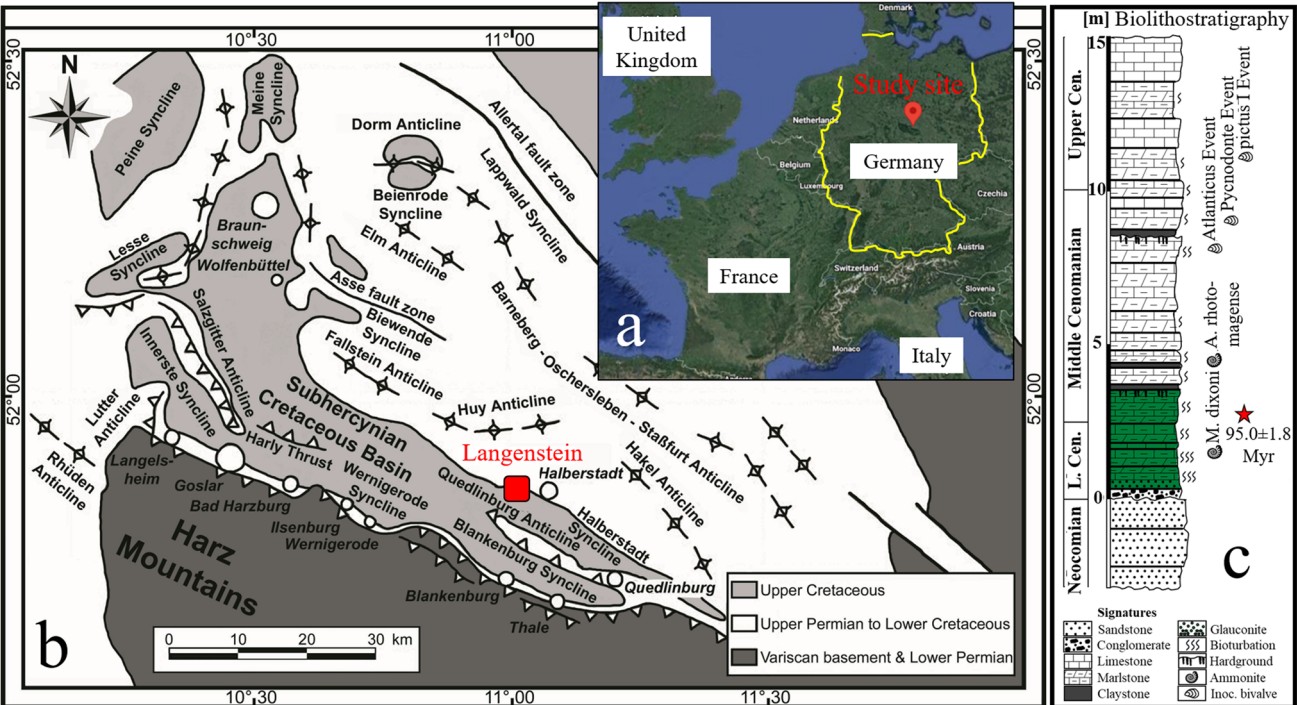

**Fig. 1 Overview of the glauconite-bearing sedimentary sequence at Langenstein. a** Satellite view of the study site in Germany (Map data ©2022 Google). **b** Geological map of the Subhercynian Cretaceous Basin with the location of the Langenstein profile. **c** Lithostratigraphic log of the Langenstein profile with the glauconite-bearing interval marked with green color. The biostratigraphy is from ref. [38], while the glauconite K-Ar age is from ref. [40] (sample position in the log is marked by the red star).

Figs. 2–4 for spatial element distribution mappings of the green grains). The variable composition of the glauconites, the absence of oxidized grain surfaces, the lobate form of the grains, the presence of cracks in the pellets, and the occurrence of glauconite infillings within chambers of foraminifera indicate the authigenic nature of the glauconites.

The plot of the chemical composition data in the $Al_2O_3$ vs TFe and $K_2O$ vs $SiO_2$ diagrams (Fig. 2d, e) indicates that glauconite formation progressed through the substitution of $Fe^{3+}$, $Fe^{2+}$, and $Mg^{2+}$ ions for $Al^{3+}$ ions in the octahedral sites and of $Al^{3+}$ ions (and eventually $Fe^{3+}$ ions) for $Si^{4+}$ ions in the tetrahedral sites. The resulting negative layer charge was balanced by the uptake of $K^+$ ions and minor $Na^+$ and $Ca^{2+}$ ions in the interlayer sites of the glauconite[47]. Glauconites preferentially develop in organic-rich, semi-confined micromilieus, such as in fecal pellets and in foraminifera chambers, close to the sediment–seawater interface through the reaction of Fe(III)-smectite precursors with mono-silicic acid, goethite (inherited from the sediment) and seawater- or pore water-derived $K^+$ and $Mg^{2+}$ ions[40]. As the mixed-layered glauconite-smectite clay transforms into glauconite, $Na^+$, $Ca^{2+}$, and hydrogen ($H^+$) ions are released from the crystal lattice, as supported by our chemical composition data (see Supplementary Table 1). This mode of glauconite formation is representative of many modern and palaeo-shelf environments[35].

**Glauconite abundance in the palaeo-depositional context.** The bottom part of the Langenstein profile contains continental sandstones, which are unconformably overlain by an inner-shelf conglomerate, ~30 cm thick, and subsequently deposited glauconite-bearing strata (see Supplementary Information for details on different lithologies, stratigraphic framework, and sample material). Two glauconite-bearing lithologies are recognizable across the profile (Fig. 3a): A sandstone bed rich in glauconite (up to ~70 wt%), ~40 cm thick, and the glauconite-

bearing carbonates (so-called Glauconitic Pläner Limestones). The limestones have a highly variable glauconite content, ranging from ~20–25 wt% at the base (1.1–1.6 m) and ~5–10 wt% in the middle part (1.6–2.5 m) to ~1 wt% at the top of the profile (2.4–3.4 m) (Fig. 3b). Hence, the glauconite-bearing interval has a cumulative thickness of ~2.8 m (sandstone plus Pläner Limestones), reflecting the estimated sediment accumulation rate of ~1.3 m Myr$^{-1}$ and the absolute duration (~1.8 Myr) of the glauconite-bearing Mantelliceras dixoni Zone[39] (henceforth called M. dixoni Zone)[48], which lasted from ~97.9 Myr to ~96.1 Myr. However, this linearized bulk sedimentation rate is much lower compared to the marine mid-shelf sequences of Northern Germany (~70 m Myr$^{-1}$ at Wunstorf)[39], which is due to the low productivity of the carbonate factory and low clastic sedimentation on the palaeo-shelf at Langenstein. Such trans-gressive systems tracts and reduced sedimentation rates favor glauconite formation and accumulation. The evolved nature and the high abundance of glauconite at the bottom part of the Langenstein profile indicate mega-condensed sedimentation, while the lower abundance of glauconite up-section in the profile suggests low to moderate sedimentation[34,49]. Similar shallow marine condensed deposits containing glauconite are reported globally from the Cenomanian[43].

The carbon and oxygen isotopic composition (see Supplementary Table 2) of calcite spar within the sandstones and the conglomerate (−7.5 to −0.8‰ of $\delta^{13}C$, VPDB, and −7.5 to −5.4‰ of $\delta^{18}O$, VPDB), as well as the positive linear correlation between the $\delta^{13}C$ and $\delta^{18}O$ values ($R^2 = 0.995$), suggest a continent-derived carbon source (i.e., low $\delta^{13}C$ values inherited from soil organic matter) and minor diagenetic overprinting (i.e., low $\delta^{18}O$ values inherited from the interaction with meteoric or burial fluids) (Fig. 3c, d). The calcite matrix within the glauconitic sandstone records the transition from continental influences to progressively more marine sedimentation (−6.0 to 0.4‰ of $\delta^{13}C$, VPDB, and −7.3 to −5.0‰ of $\delta^{18}O$, VPDB), while the carbonate mud within the Glauconitic Pläner Limestones

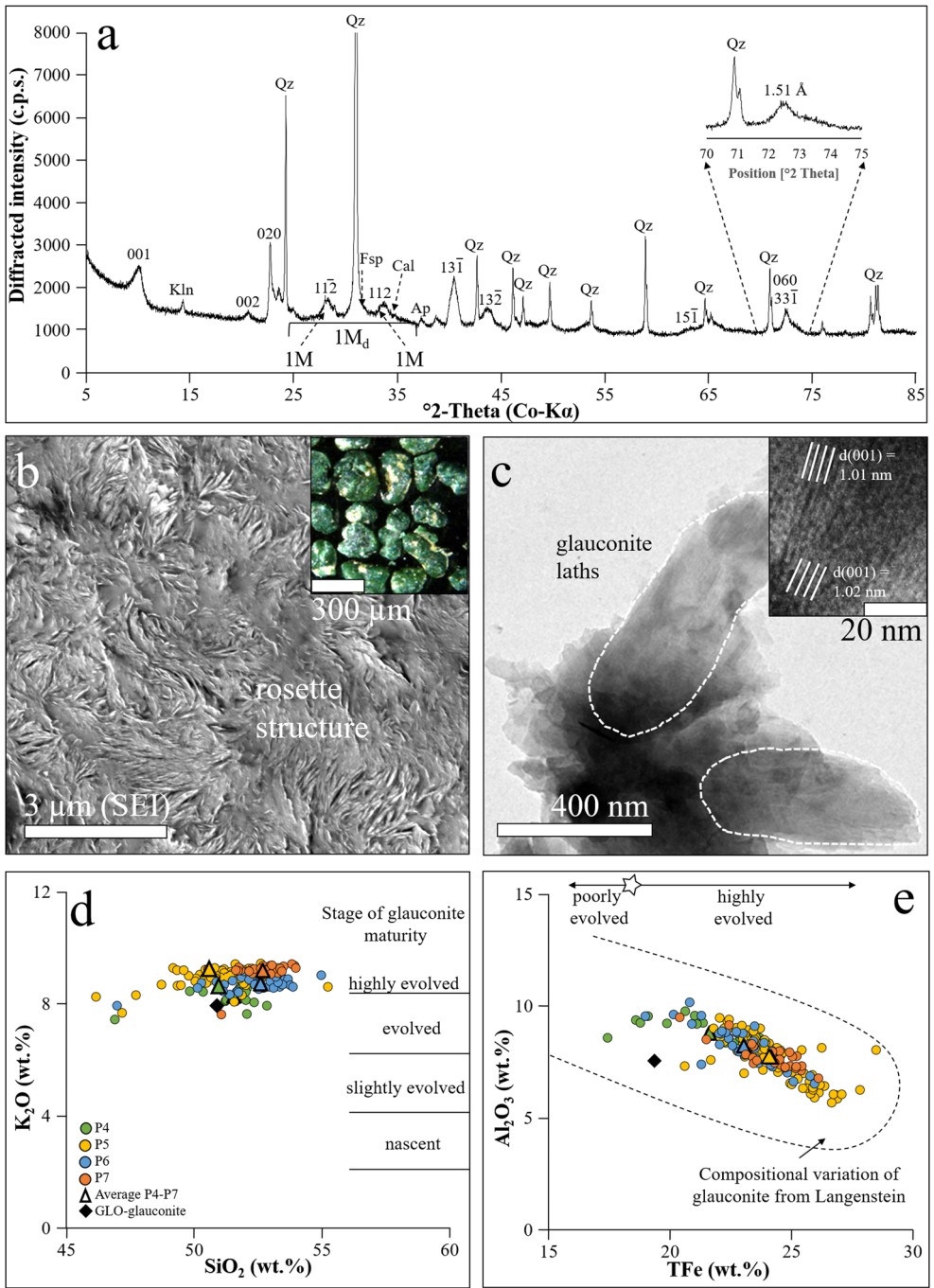

**Fig. 2 Glauconite characterization. a** The XRD pattern of the glauconite-bearing sandstone identifies ordered glauconite (1 M polytype), disordered glauconite-smectite (1 M$_d$ polytype), quartz (Qz) and minor kaolinite (Kln), feldspar (Fsp), apatite (Ap), and calcite (Cal). **b** The secondary electron image (SEI) shows "rosette" structures within the dark green fecal pellets (cf. inset), which is typical for "evolved" glauconite grains. **c** The TEM images highlight the majority of 10 Å domains (cf. inset) within the lath-like glauconite particles. **d, e** The chemical composition of the glauconites records "evolved" to "highly evolved" and Fe-rich grains, which is representative of Mesozoic to Cenozoic glauconites.

exhibits isotopic signatures typical for shallow marine carbonate sedimentation (1.0 to 2.1‰ δ¹³C, VPDB, and −3.9 to −3.4‰ of δ¹⁸O, VPDB) (Fig. 3c, d). Calcite mud precipitation within the glauconite-bearing interval occurred at a temperature of around 26 ± 2 °C, which is typical of a "warm" shelf environment[50]. Thus, the palaeo-depositional setting, as well as the composition and the mode of glauconite formation at Langenstein are representative of many modern and palaeo-shelf environments, justifying the use of the Langenstein section for the calculation of rates and fluxes for global shelf settings.

**Rate of elemental uptake by glauconite in marine settings**. Reverse weathering reactions to produce authigenic clay minerals can significantly impact the ratio of element diffusive return fluxes to seawater (i.e., recycling) vs element sequestration in marine sediments. Previous studies have identified "hot spot" areas that favor clay mineral precipitation in marine sediments, such as mangrove forests, deltas, and estuaries[21,24,26,27,51], as well as low- to high-temperature hydrothermal sites[4–7], and shallow-water settings characterized by reduced sedimentation[34,46]. In such surroundings, clay retrograde reactions are important controls on marine elemental

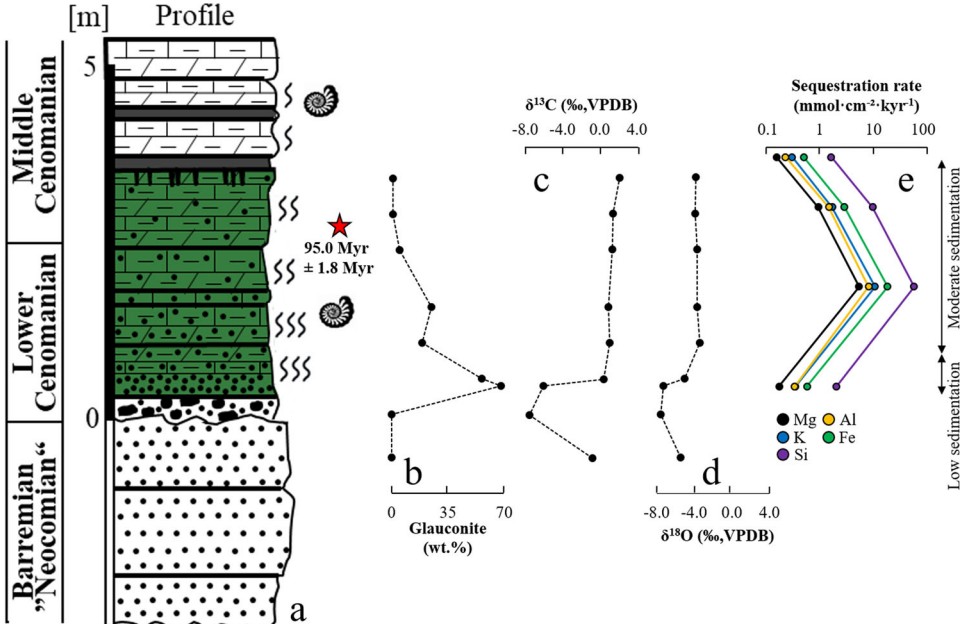

**Fig. 3 Glauconite abundance, δ¹³C and δ¹⁸O isotope profiles of calcite and major element sequestration rate related to glauconite formation. a** Lithostratigraphic log of the studied profile with the glauconitized interval highlighted in green color (refer to Fig. 1c for legend). **b** Glauconite abundance across the Langenstein profile. **c, d** Carbon and oxygen isotopic profiles of calcite indicate the glauconitized interval to be deposited on a warm proximal shelf environment. **e** Major element sequestration rates attributed to glauconite formation.

output fluxes at or close to the sediment-seawater interface. Here, we constrain glauconite-associated sequestration rates for two classes of major elements, i.e., (i) "conservative elements" (K and Mg), which have high concentrations in seawater (hundreds of ppm) and long residence times (few Myr), and (ii) "scavenged elements" (Al, Si, and Fe), which are depleted in present-day seawater (sub-ppm levels) and have short residence times (few kyr). During glauconite formation, K and Mg are believed to be primarily sourced from seawater or seawater-derived pore fluids, while Al, Si, and Fe are mostly derived from marine sediment sources via dissolution processes of existing mineral phases (e.g., decay of lithogenic particles and biogenic silica, and reductive dissolution of Fe-(hydr)oxides)[47]. We do not consider Na and Ca, as they are barely incorporated in glauconite (see Supplementary Table 1).

Assuming a sedimentation rate of 0.13 cm kyr⁻¹ for the basal part of the M. dixoni Zone and of 7.0 cm kyr⁻¹ for the upper part of the M. dixoni Zone (Fig. 3e)[39], as well as a sediment density of 2.7 g cm⁻³ for the glauconitized strata, and considering the composition of the glauconites (see Supplementary Table 1) and the abundance of glauconite across the Langenstein sequence (Fig. 3a, b), element-specific sequestration rates associated with glauconite formation can be calculated (Fig. 3e and Supplementary Table 3). The element sequestration rates range from 0.4 to 8.9 mmol K cm⁻² kyr⁻¹, 0.2 to 4.9 mmol Mg cm⁻² kyr⁻¹, 0.3 to 7.1 mmol Al cm⁻² kyr⁻¹, 1.7 to 40.3 mmol Si cm⁻² kyr⁻¹ and 0.6 to 14.6 mmol Fe cm⁻² kyr⁻¹, reflecting the different sedimentation rates and the extremely low to very high abundances (1 vs 70 wt%) of glauconite in the profile.

Considering an average glauconite content of 5–10 wt% for the Langenstein sequence, which is representative of Phanerozoic glauconite deposits (7 ± 4 wt%)[35], the glauconite-associated elemental uptake rate averages ~2.6 ± 1.2 mmol K cm⁻² kyr⁻¹, ~1.5 ± 0.7 mmol Mg cm⁻² kyr⁻¹, ~2.2 ± 1.1 mmol Al cm⁻² kyr⁻¹, ~12.3 ± 5.8 mmol Si cm⁻² kyr⁻¹, and ~4.2 ± 2.0 mmol Fe cm⁻² kyr⁻¹. For comparison, the sequestration rates for glauconite-smectite and glauconite forming in the modern deep-sea sediments of the Ivory Coast (Ghana Marginal Ridge) were determined as 20 µmol K cm⁻² kyr⁻¹, 30 µmol Mg cm⁻² kyr⁻¹, 30 µmol Al cm⁻² kyr⁻¹, 250 µmol Si cm⁻² kyr⁻¹, and 80 µmol Fe cm⁻² kyr⁻¹ (partly recalculated from ref. [28]), which is, on average, ~50–130-times lower than the Langenstein sequestration rate. This is because the glauconite content (2.5 wt%, on average), the K concentration (2.9 wt%, on average, reflecting the "nascent" stage, according to the glauconite maturity classification of ref. [33]) and the rate of sedimentation (~5-times lower) are substantially lower in deep-water settings than in the shelf regions. The slower rate of glauconite formation in deep-water settings is mainly due to the low temperature (~5 vs ~25 °C) and the reduced supply and sedimentary reflux of Al³⁺ ions and organic matter. The latter controls local redox restrictions (semi-confined micromilieu vs redoxcline) that predetermine the availability and the speciation of Fe (Fe²⁺ vs Fe³⁺), which is the rate-determining factor for glauconite formation[36].

**Elemental burial by glauconite in modern marine settings**. To the best of our knowledge, elemental output fluxes attributed to widespread glauconite formation taking place at the shallow shelf (defined here as 0–200 m water depth) and in the deep-sea (defined here as >2000 m water depth) of the modern oceans have not been determined yet and are not fully accounted for in earth system models[8,9,14,21,24,25,28,51]. We recognize that some of the glauconite deposits of the Quaternary and Holocene are of para-autochthonous or detrital origin, representing reworked glauconites of the Neogene or older age[33]. As a first-order approximation to calculate the present-day major element output fluxes attributed to green clay authigenesis, we use published glauconite contents (5.6 wt% vs 2.5 wt%)[35,52] and compositions (K: 7 vs 3 wt%, Mg: 3 vs 2 wt%, Al: 4 vs 4 wt%, each ±1 wt%, Si: 24 vs 28 wt%, Fe: 18 vs 18 wt%, each ±2 wt%)[28,35] in the shallow and deep marine sediments of the Holocene, the total areas of the modern shelf and deep-sea regions (27.12 × 10¹² m² vs 302.5 × 10¹² m²)[18], an average sediment density of 2.7 g cm⁻³, and estimated global sedimentation rates for shallow-water vs deep-water settings (10–20 cm kyr⁻¹ vs

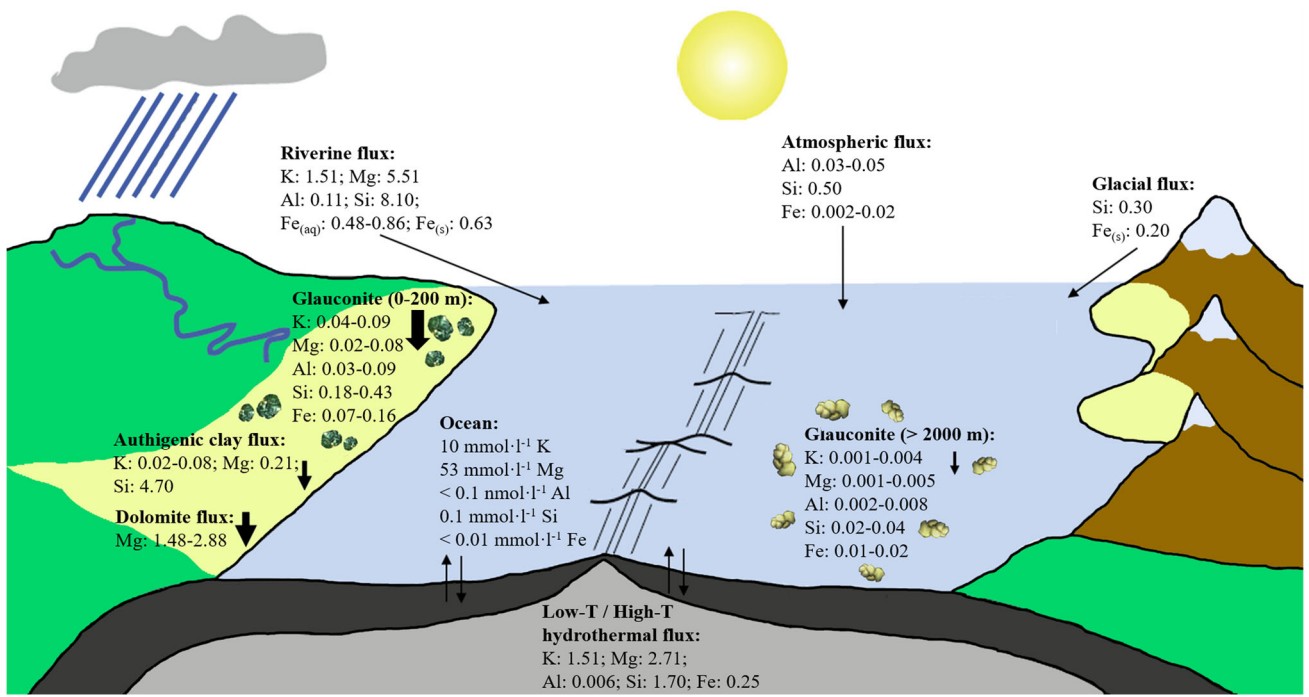

**Fig. 4 Simplified sketch of the global K, Mg, Al, Si, and Fe cycle with average fluxes of the main oceanic sources and sinks (in Tmol·yr⁻¹).** The directions of the arrows refer to the net uptake or release of elements for each geological process. Elemental fluxes due to phosphate and evaporate formation and dissolution are not considered. K fluxes: refs. [24, 56, 58–60]; Mg fluxes: refs. [4, 6, 56, 61]; Al fluxes: refs. [62, 63]; Si fluxes: ref. [51]; Fe fluxes (aq – aqueous/dissolved; s – solid/particulate): refs. [1, 18]; Glauconite fluxes: ref. [28], this study; Ocean chemistry: refs. [4, 56].

0.4–0.8 cm kyr⁻¹)[53,54] (Supplementary Table 4). Upscaling predicts global output fluxes associated with glauconite formation of ~0.04–0.09 Tmol K yr⁻¹, ~0.02–0.08 Tmol Mg yr⁻¹, ~0.03–0.09 Tmol Al yr⁻¹, ~0.18–0.43 ;Tmol Si yr⁻¹, and ~0.07–0.16 Tmol Fe yr⁻¹ at the shallow shelf, and of ~0.001–0.004 Tmol K yr⁻¹, ~0.001–0.005 Tmol Mg yr⁻¹, ~0.002–0.008 Tmol Al yr⁻¹, 0.02–0.04 Tmol Si yr⁻¹, and ~0.01–0.02 Tmol Fe yr⁻¹ for the deep-sea glauconites (Fig. 4). The calculated elemental fluxes have relatively high uncertainty but are well within global marine fluxes published in the literature[4,6,18,51,55,56]. However, we consider these fluxes to be conservative estimates, given that they are calculated assuming an overall low to moderate sedimentation rate (i.e., present-day shelf areas display sedimentation rates between 0.1 and 1.0 cm yr⁻¹)[57], which is often associated with glauconite formation in modern shelfal sediments[33,47,49].

Deep-water glauconite formation is not significant in the context of marine K budgets, accounting for the removal of merely <0.3 % of the total dissolved riverine K influx (~1.51 Tmol yr⁻¹)[58], and of the K supplied to the ocean by hydrothermal alteration of the modern oceanic crust (~1.51 Tmol yr⁻¹)[59]. Shallow-water glauconite formation, by contrast, can play an important role in the global K cycle (Fig. 4), sequestering ~3–6% of the total oceanic K inventory that is sourced from riverine and hydrothermal fluxes or ~2–5% of the K that is removed from the ocean via low-temperature basalt alteration (~1.99 Tmol yr⁻¹)[56,60]. Hence, K sequestration by glauconite formation at the shelf is at the same order of magnitude as K burial by authigenic Fe-illite formation taking place in the mangrove forests worldwide (~0.02–0.08 Tmol yr⁻¹)[24], so that changes to K uptake rates by green clay authigenesis have the potential to significantly alter seawater composition over time.

The same conclusions can be drawn for the marine Mg cycle (Fig. 4): Glauconite formation at the shelf consumes <2% of the terrestrial Mg flux (~5.51 Tmol yr⁻¹) that is brought to the ocean

via continental weathering of Mg-bearing carbonates and silicates and constitutes ~1–3% of the marine Mg sink that is associated with oceanic crust alteration (~2.71 Tmol yr⁻¹)[4]. Further, Mg sequestration by glauconite is equivalent to ~1–5% of the estimated Mg sink by enigmatic (yet hidden) dolomite deposits (~1.48–2.88 Tmol yr⁻¹)[6] or ~10–38% of the Mg consumption by authigenic clays forming in the Amazon deltaic sediments (~0.21 Tmol yr⁻¹)[56]. Contrary, Mg sequestration associated with deep-water glauconite formation accounts for only <1% of the low-temperature alteration flux (~0.66 Tmol yr⁻¹)[6,56,61].

As for the Al cycle (Fig. 4), shallow-water glauconite formation is significant, contributing to ~27–82% removal of the dissolved riverine Al influx to the oceans (~0.11 Tmol yr⁻¹)[62], which is consistent with the estimated high loss of dissolved Al to estuarine and shelfal sediments. Deep-water glauconite formation is also critical in the context of marine Al budgets, accounting for ~4–27% loss of the marine Al inventory related to atmospheric dust deposition (~0.03–0.05 Tmol yr⁻¹)[63] or almost complete removal of the Al flux injected from hydrothermal vents (~0.006 Tmol yr⁻¹)[63]. Thus, we propose that Al uptake via green clay authigenesis could act as an important (yet overlooked) Al sink in deep marine sediments, where active reversible (adsorptive) scavenging of Al in the water column, as well as Al incorporation into diatoms, are currently thought to take key control on the vertical flux and recycling of Al[63].

For a long time, reverse weathering reactions at the sediment-seawater interface were thought to constitute only a minor sink of Si in the global ocean (~0.03–0.6 Tmol yr⁻¹)[64]. However, extrapolation of cosmogenic ³²Si data obtained from tropical and subtropical deltas suggests that ~4.7 (±2.3) Tmol yr⁻¹ of Si is incorporated into authigenic clays on a global scale[26,51]. The Si flux related to glauconite formation is ~0.18–0.43 Tmol yr⁻¹ for the shelfal areas and ~0.02–0.04 Tmol yr⁻¹ for the deep-sea, respectively (Fig. 4), which indicates that this process removes

~2–5% of the riverine Si influx to the ocean (~8.1 Tmol yr$^{-1}$) or ~1–2% of the hydrothermal Si flux (~1.7 Tmol yr$^{-1}$), 4–8% of the atmospheric Si flux (~0.5 Tmol yr$^{-1}$), and 5–10% of the glacial Si flux (~0.4 Tmol yr$^{-1}$)[51]. Nevertheless, we note that global marine gross Si bio-productivity, mostly due to silicifying algae such as diatoms, is estimated at ~255 (±52) Tmol yr$^{-1}$, representing the first-order control on the modern marine Si cycle[51].

Glauconite acts as an important sink for Fe (Fig. 4), with shallow-water glauconite formation accounting for up to ~8–33% removal of the dissolved and particulate riverine flux of highly reactive Fe to the ocean (~0.48–0.86 Tmol yr$^{-1}$ and ~0.63 Tmol yr$^{-1}$)[1]. Although Fe uptake by deep-water glauconite formation is less significant (Fig. 4), it is still at the same scale as the hydrothermal alteration flux (~0.25 Tmol yr$^{-1}$) or the glacial (~0.20 Tmol yr$^{-1}$) and atmospheric dust fluxes (~0.02 Tmol yr$^{-1}$)[1]. Even though oxidation and scavenging processes are the first-order controls on the benthic Fe fluxes in the ocean[18], we argue that glauconite formation in shallow and deep marine settings is important, but currently underestimated Fe sink.

**Effect of glauconite on global marine palaeo-fluxes**. The source-sink relations of chemical elements in the modern ocean are increasingly well constrained[1,2,4,6,14,18,21,24,51,65,66], but the palaeo-fluxes related to authigenic clay formation remain enigmatic. The element uptake rates reported for glauconite formation at Langenstein (shallow-water; this study) and the Ivory Coast (deep-water; recalculated here from data reported in ref. [28]) may not be directly transferrable to all other marine settings that accumulated glauconite through time and space. However, the mode of glauconite formation (Fe-smectite-to-glauconite reaction), the micro-environment (fecal pellets and foraminifera chambers), the timing (<1 Myr), the composition (Fe-rich), the abundance (5–10 wt% vs 2–3 wt%), and the depositional environment (warm shallow shelf vs cool deep-sea) at the two localities are representative of the range expected for many modern and past glauconite-forming environments[28,33,35,36,40,42–45,47]. The Langenstein glauconites share similarities with other Mesozoic to Cenozoic glauconite deposits, such as a similar abundance (7 ± 4 wt% in the marine rock record from the Triassic to the Holocene; Fig. 5a)[35] and comparable chemical composition (7 ± 1 wt% K, 3 ± 1 wt% Mg, 4 ± 1 wt% Al, 24 ± 2 wt% Si, and 18 ± 2 wt% Fe; Fig. 5b)[35]. If we assume a sediment density of 2.7 g cm$^{-3}$ and variable sedimentation rates between 0.1 and 100 cm kyr$^{-1}$ (Fig. 5c)[53,54] are representative of the global shelf through geological time, we can compute major element palaeo-sequestration rates for shallow-water glauconite formation for Mesozoic and Cenozoic times (Fig. 5d–h and Supplementary Table 4). With the recognition of the aforementioned global elemental fluxes of the modern ocean, we propose that the obtained "low" and "high" palaeo-sequestration rates are underestimated (i.e., ≤1 cm kyr$^{-1}$; functionally zero rates at the yearly time scale are typical for deep oceanic basins) and overestimated (i.e., ≥100 cm kyr$^{-1}$; such rates are typical for continental margins associated with major rivers, deltaic sediments, upwelling zones, and geologically young glacial deposits), respectively[53,54,57]. Thus, the "moderate" sedimentation rate (~10 cm kyr$^{-1}$; this rate is comparable with authigenic illite formation in mangrove forests)[24] better represents major element uptake related to green clay authigenesis.

It is evident that glauconite formation significantly contributed to major element sequestration in shallow marine sediments throughout Earth's history, averaging 3 ± 2 mmol K cm$^{-2}$ kyr$^{-1}$, 2 ± 1 mmol Mg cm$^{-2}$ kyr$^{-1}$, 3 ± 2 mmol Al cm$^{-2}$ kyr$^{-1}$, 16 ± 9 mmol Si cm$^{-2}$ kyr$^{-1}$, and 6 ± 3 mmol Fe cm$^{-2}$ kyr$^{-1}$, respectively. We note that major element sequestration by glauconite formation also occurred in the older sediments of the Archean, Proterozoic, and early Cambrian, but

this elemental uptake cannot be adequately quantified, given that sedimentary archives of this time are scarce and that most of the old glauconites are at least partly altered to illite or chlorite minerals[44]. We are, however, able to make some general inferences. Element sequestration by green clay authigenesis was likely of minor importance in the Late Ordovician, Early Silurian, and Late Devonian, which corresponded to major glacial events, when glauconite formation is inefficient[35]. Conversely, glauconite formation and elemental uptake is likely to have been favored at other times in the Earth's past, e.g., during intervals of extensive marine anoxia, which featured an elevated seawater Fe pool compared to the present, or before the advent and global expansion of marine pelagic silicifiers (i.e., sponges, radiolarians, diatoms) decreased the seawater Si reservoir from ~550 Myr onward[67,68]. We, therefore, infer that these elements may have been sourced from seawater rather than from the dissolution of Fe-(hydr)oxides, biogenic silica, or clastic silicates during these times. Returning now to the Mesozoic and Cenozoic, we find that high glauconite abundances and related high element sequestration rates are evident, for example, in the Neogene (glauconite sands from the Chatham Rise, Southwest Pacific), at the Paleogene-Eocene Transition (glauconite sands from the continental margins of the northern hemisphere; Upper and Lower Greensands of England) and in the Cretaceous (New Jersey, Maryland and Delaware Greensands; greensand giants from the Duwi group, Egypt; Bakchar glauconite deposit, Western Siberia) (Fig. 5d–h). These periods record glauconite deposits of huge economic or geological value, with K and Mg being mainly sourced from seawater and Al, Si, and Fe being mostly inherited from the marine sediments.

Using average elemental sequestration rates per geological period (Fig. 6a) and corresponding occurrences of glauconite on the shelf (Fig. 5a), as well as calculated low and high estimates of the shallow ocean areas over time (Fig. 6b)[69,70], we can compute major element palaeo-fluxes (Tmol yr$^{-1}$) associated with green clay authigenesis that progressed on the world's shelf area over time (Fig. 6c–g and Supplementary Table 5). Based on comparison with global major element fluxes of the modern and past ocean, we propose that the obtained "high" palaeo-fluxes are overestimated (i.e., the shelf areas reported by ref. [69]) and that the "low" palaeo-fluxes (i.e., the shelf areas reported by ref. [70]) better portray the average elemental burial related to green clay authigenesis per geological period, which averages 0.07 ± 0.09 Tmol K yr$^{-1}$, 0.05 ± 0.06 Tmol Mg yr$^{-1}$, 0.05 ± 0.08 Tmol Al yr$^{-1}$, 0.32 ± 0.44 Tmol Si yr$^{-1}$, and 0.12 ± 0.17 Tmol Fe yr$^{-1}$ during the Triassic to Holocene.

The ratio of the elemental palaeo-fluxes associated with glauconite formation in the past vs modern ocean (Fig. 6h) indicates further that the elemental fluxes were much higher from the Jurassic to the Oligocene compared to the modern ocean, averaging a factor of 2.1 ± 1.7, which we attribute to the warm and sea level highstand "greenhouse" conditions prior to the Eocene-Oligocene transition. The lower elemental palaeo-fluxes ever since the Oligocene are caused by the decrease of the shallow-water shelf areas and seawater temperature with the onset of the first Southern Hemisphere glaciation (~34 Myr ago) and then Northern Hemisphere glaciation (~5 Myr ago), where glauconite formation is reduced. Although these estimates have relatively high uncertainty and need to be better constrained in future work, it is evident that green clay authigenesis greatly affected the global marine element cycles throughout Earth's history.

We conclude that fast retrograde clay mineral reactions, which occur widely on the ocean floor, are of great significance to the marine element cycles and have to be considered in present and past earth system models. The major element burial fluxes attributed to green clay authigenesis were significantly higher under sea level highstand and "greenhouse" conditions in the majority of the Phanerozoic compared to the modern sea level lowstand and "icehouse" conditions, which suppress glauconite formation. It is now

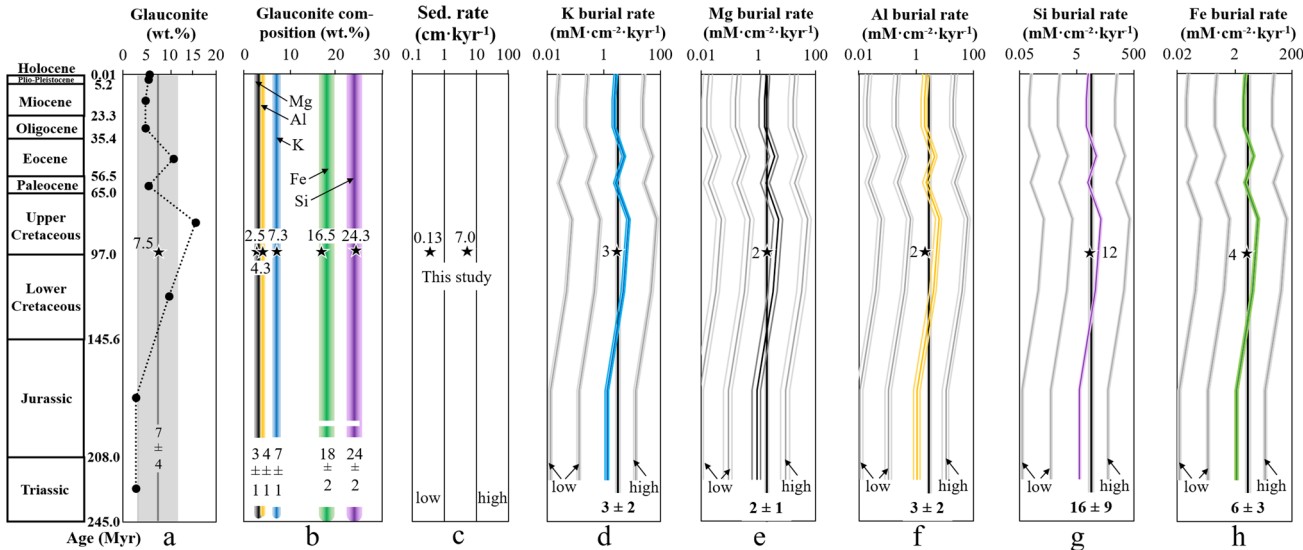

**Fig. 5 Major element sequestration rates for shallow-water glauconite formation during the Mesozoic and Cenozoic. a** Glauconite abundance on the shelf through time[35]. **b** Average composition (±2 SD) of glauconite over time[35]. **c** Variation in overall sedimentation rate typical for shelf areas[53, 54]. **d–h** Elemental sequestration rates associated with shallow-water glauconite formation through time (colored curves) and at Langenstein (black stars). Average sequestration rates (±2 SD) are highlighted by the gray shaded intervals. The calculations are based on constant sedimentation rates and a sediment density of 2.7 g cm$^{-3}$ for each geological period.

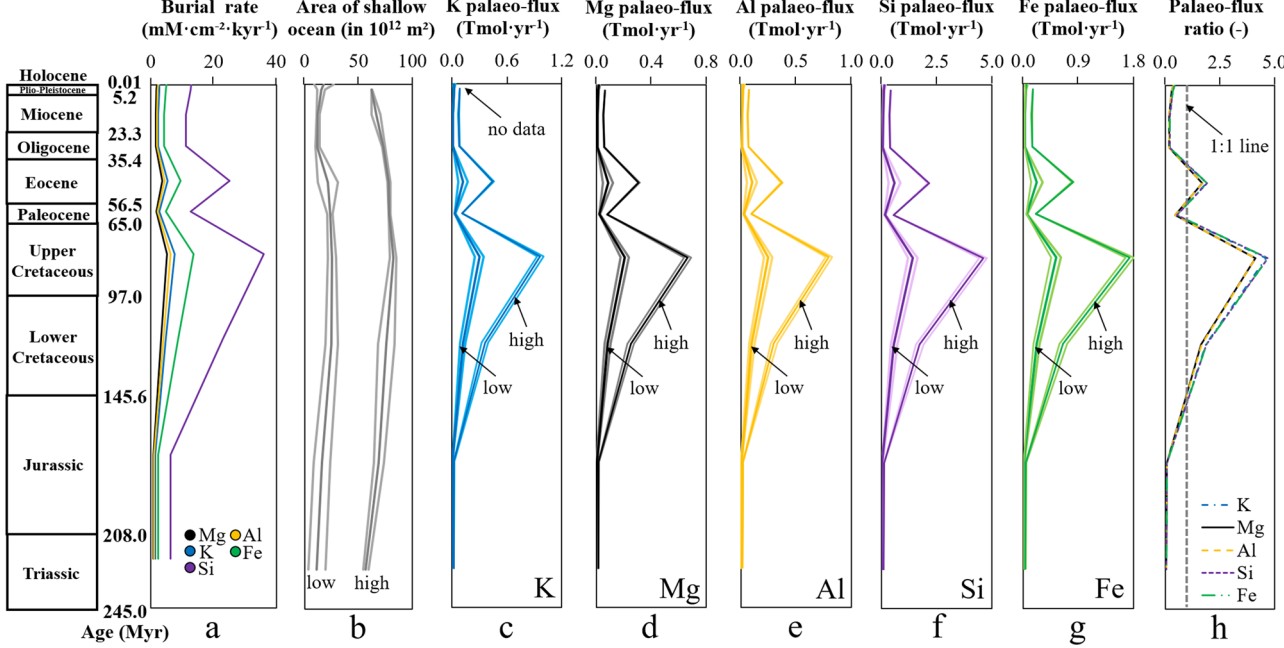

**Fig. 6 Element palaeo-fluxes (Tmol yr$^{-1}$) associated with green clay authigenesis on the shelf during the Mesozoic and Cenozoic. a** Average major element burial rates associated with shallow-water glauconite formation through time (see Fig. 5d–h). **b** Area of the global shallow ocean over time-based on two different paleogeographic reconstructions. The estimate labeled as "low" is the shelf area (average ± 2 SD) that was computed from the respective contours of a series of palaeo-digital elevation models (1° × 1° resolution) developed by ref. [70] using Generic Mapping Tools[74]. The estimate labeled as "high" is the shelf area (average ± 2 SD), which was calculated by ref. [69] based on the paleogeographic model from ref. [75] that is available as a series of polygons, reconstructed with the plate tectonic model of ref. [76]. **c–g** Element palaeo-fluxes (average ± 2 SD) associated with shallow-water glauconite formation for the "low" vs "high" paleogeographic reconstructions. **h** Element palaeo-flux ratios (ancient vs modern ocean) for the "low" shallow ocean scenario. The palaeo-flux calculations assume a constant occurrence of glauconite (see Fig. 5a) for each geological period.

up to future studies to estimate how element sequestration through glauconite formation impacts the isotopic composition of the ocean, the pore water reservoir and the marine sediments, and to assess the impact of climate change through time on the elemental burial fluxes attributed to green clay authigenesis.

## Methods

X-ray diffraction (XRD) patterns were recorded on powdered bulk rocks using a PANalytical X'Pert PRO diffractometer equipped with a high-speed Scientific X'Celerator detector and operated at 40 kV and 40 mA (Co-Kα radiation source). The samples were prepared using the top-loading technique[71]. The preparations were examined in the range 4 to 85° 2θ with a step size of 0.008° 2θ and a scan

speed of 40 s. Mineral quantification was carried out by Rietveld analysis of the XRD patterns using the PANalytical X'Pert HighScore Plus Software and the ICSD database[71]. The analytical error is better than ±3 wt%[41].

The micro-texture and the chemical composition of the green grains were analyzed on polished thick sections by electron microprobe analyses (EMPA) using a JEOL JXA8530F Plus Hyper Probe at Karl-Franzens-University Graz. Analytical conditions were 15 keV accelerating voltage, 15 nA beam current, and defocused beam, ~3 μm in size, to avoid mineral damage during the measurements. The chemical data were standardized against a range of natural and synthetic crystals, which included the following elements with their characteristic spectral lines: Al-Kα, Si-Kα, and K-Kα (microcline), Mg-Kα and Ca-Kα (augite), Fe-Kα (ilmenite), Na-Kα (tugtupite), and P-Kα (LaPO$_4$). Counting times were set to 10 s on peak and 5 s on background-position on each side of the element-specific peak. Only compositions with an analytical error of less than 7 wt% off 100 wt% were taken into further consideration (Supplementary Table 1). The chemical compositions were corrected for the average Fe(II)/Fe(III) ratio of the green grains reported by ref. [40] based on electron energy-loss spectroscopy (EELS) analyses. Structural formulae were calculated based on 22 negative charges, assuming (i) $^{IV}Si^{4+} + {}^{IV}Al^{3+}$ is equal to 4, (ii) $Fe^{2+/3+}$, $Mg^{2+}$, and $Al^{3+}_{rest}$ occupy the octahedral sheet, (iii) $K^+$, $Na^+$, and $Ca^{2+}$ are located within the interlayer sites and (iv) the P$_2$O$_5$ contents belong to apatite impurities and not to glauconite. Furthermore, element distribution maps (Al, Fe, K, Si, Ca, Mg, F, Na, S, and P) of 1200 × 1200 pixel resolution were acquired (Supplementary Figs. 2–4). The analytical conditions were as follows: focused beam, 15 keV accelerating voltage, 20 nA beam current, 3 μm pixel size, and a dwell time of 13 ms step$^{-1}$.

The particle form and the nature of the green grains were determined by transmission electron microscopy (TEM) using an FEI Tecnai F20 instrument operated at an accelerating voltage of 200 kV and fitted with a Schottky field emitter, a Gatan imaging filter, and an UltraScan CCD camera. High-resolution TEM lattice fringe images were collected parallel to the (001)-plane of the clay minerals particles. Therefore, a sub-fraction of the Glauconitic Pläner Limestones was treated with 10% acetic acid for 1 h to dissolve carbonates. The acid-insoluble residue was washed several times with ultrapure water and subsequently, the green grains were separated by hand-picking under a binocular microscope. The green grains were ultrasonically dispersed for 15 min in ethanol and prepared following standard procedures prior to the TEM analyses[71].

Stable carbon and oxygen isotope compositions of carbonates with calcite mineralogy only were obtained on the evolved CO$_2$ after the reaction of powdered whole-rock sub-samples with pure H$_3$PO$_4$ at 72 °C in an automated GASBENCH II preparation unit connected to a Thermo Finnigan Delta plus XP mass spectrometer at the Institute for Geological and Geochemical Research (IGGR) in Budapest, Hungary). The isotopic values are expressed as δ$^{13}$C and δ$^{18}$O (in ‰) relative to the Vienna Pee-Dee Belemnite (VPDB) reference material (Supplementary Table 2). On the basis of standard measurement, the accuracies of the δ$^{13}$C and δ$^{18}$O values are estimated to be better than ±0.1‰[72]. Calcite formation temperatures (in °C) were calculated exclusively for the Glauconitic Pläner Limestones based on the measured δ$^{18}$O values assuming a δ$^{18}$O value of −1‰ (VSMOW) for Cretaceous seawater[73].

## Data availability

The data generated in this study can be accessed via the Zenodo Data Repository (https://doi.org/10.5281/zenodo.5994622). All raw data are also provided in the Supplement. The studied geological samples from the Langenstein section are archived in the mineral collection of the Institute of Applied Geosciences (Graz University of Technology) and can be made available upon request to A.B. or M.D. All samples were collected and exported in a responsible manner, and in accordance with relevant permits and local laws.

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

## Acknowledgements

This work was partly supported via the ARC Discovery Project (DP210100462; grant to J.F.) tilted "Glauconite: Archive recording the timing and triggers of Cambrian radiation" and the NAWI Graz Geocenter. We acknowledge Cyrill Grengg (TU Graz) for assistance with the EMPA.

## Author contributions

A.B. designed the study. A.B., S.B., S.L., U.M., E.S., and T.Z. participated in the early-stage discussion. G.C. provided the $\delta^{13}C$ and $\delta^{18}O$ isotopic data and N.M.W. contributed to the areas of the global shallow ocean. A.B., M.D., S.L., and J.F. calculated the K–Mg–Fe sequestration rates and the K–Mg–Fe palaeo-fluxes. A.B., U.M., E.S., and T.Z. carried out the fieldwork. All coauthors contributed to the writing.

## Competing interests

The authors declare no competing interests.
