## [Peer Review File · Nature Communications]

Title: Impact of green clay authigenesis on element sequestration in marine settingsREVIEWER COMMENTS

Reviewer #1 (Remarks to the Author):

In this manuscript, the authors present a new suite of sedimentological and geochemical data characterizing the composition and abundance of glauconite in a section of Cretaceous shelfal sedimentary strata in Germany. They combine this new data with previously-published information to extrapolate to the fluxes of K, Mg, and Fe into glauconite across Triassic through Holocene time and argue that authigenic glauconite formation has been a significant sink across this time period. In general, clays formed through reverse-weathering type reactions are (rightfully!) being increasingly recognized as important contributors to many elemental cycles in both the modern and deep time, so from that perspective I think this manuscript would generally be of broad interest to geochemists. However, I noticed some issues in the manuscript that merit revision prior to publication. Two of these I would consider major issues, and the rest are minor/moderate details; I describe these concerns below:

In general, the manuscript is heavy in self-citations (~20% of the citations were to previous work with Baldermann and/or Banerjee as lead or coauthor, but these citations are also the most heavily cited in the text). While I recognize that this degree of self-citation is often appropriate, particularly since both these authors have clearly conducted substantial previous work on the topic of glauconite formation, I questioned whether all these citations were necessary, and it was hard to tell how much of the work presented here was new, vs. reiterating conclusions already made in these previous publications. Furthermore, I think the heavy self-citation led to the authors incorrectly assuming some background and terminology would be familiar to readers (e.g. “evolved” and “highly evolved” glauconite), but it is really asking a lot of readers to expect that they go back and read all of the authors previous work to catch up on this terminology and context. If all these citations are truly needed, then the manuscript would benefit from the authors more clearly introducing terminology from previous works and more clearly differentiating which parts of the conclusions are novel. Finally, while the authors were liberal with self-citations, the manuscript was missing several key citations to important previous work on reverse weathering, in particular work by Rahman and colleagues using ^{32}Si to measure rates of reverse weathering, and the recent re-estimate of the marine silica cycle budget by Treguer et al. (2021). Not only were these citations missing, but the authors’ apparent unfamiliarity on important recent work on reverse weathering from the perspective of the silica cycle led them to make several factually incorrect statements (detailed below). I can mainly comment here on these missing or incorrect references with respect to Si because I am most familiar with it, but I wondered whether there might also be other key references missing in other parts of the paper. I’d suggest the authors should place more emphasis on connecting their work to studies by researchers outside the coauthor team.

I also found the organization and logic of the discussion to be counterintuitive – the authors first extrapolate from a relatively small set of data points to make estimates of elemental sink sizes for the entire Mesozoic-Cenozoic time periods, then refocus on the modern system (which appears to be mainly recapitulating data from previous studies) to justify that these fluxes are important in the modern. It seems to me that a more logical approach would address the modern, which is theoretically better

constrained first (although this is still extrapolated from only two modern sites that were presumably chosen because they have glauconite, so I question whether these are truly representative). Philosophically, the whole paper felt set up to inevitably lead to the conclusion that this process is important and didn't seem to be really set up to have the option to falsify the hypothesis that glauconite formation is significant.

Specific comments:

Lines 54-61: this sentence is extremely long and was impossible to follow given the complex grammatical structure. Please split up into multiple shorter sentences.

Lines 69-70: This could use some clarification - really excellent recent work by Rahman and others has made huge steps in better constraining reverse weathering fluxes (primarily on continental margins) which has essentially balanced the global marine Si budget (see Treguer et al 2021). So in that way, the magnitude is not "unknown". Are the authors referring more specifically to deep sea environments?

Line 74: here and in the discussion, please provide more context about these numbers. It's not entirely clear why a timescale unit is used to refer to something that has a rate.

Lines 73-82: This paragraph is missing citations to the work of Rahman et al. using the ^{32}Si tracer to characterize these rates.

Lines 96-97: Include the age of this specific section here in the text.

Line 130 & Line 134: Please define the terms "evolved" and "highly evolved" glauconite as readers of this journal are unlikely to be familiar with them.

Lines 211-216: Another area missing key references to other work while also citing multiple papers from the co-author team.

Line 225: this statement is factually incorrect about Si. Modern seawater productivity is closely tied to diatoms, who are using Si in seawater. It's not absent! Also, the most current global marine silica cycle model (Treguer et al. 2021) indicates that reverse weathering is the 2nd largest sink for Si in seawater, so it's clearly important. Equating the relatively low concentration of Si in seawater to its relative unimportance in elemental cycling or with respect to reverse weathering is not correct and therefore not adequate justification to ignore Si in the analysis.

Line 259: I'm confused by the reference to lower dissolved silica concentration in deep water settings because it is typically highest in deep waters as sinking diatom frustules dissolve.

Lines 265-268: The recent Treguer et al 2021 Si budget should be cited here.

Lines 297 and 299: why are "older" and "old" in quotations?

Figure 1: Text in panels A and C is illegibly small.

Reviewer #1 (Remarks to the Author):

In this manuscript, the authors present a new suite of sedimentological and geochemical data characterizing the composition and abundance of glauconite in a section of Cretaceous shelfal sedimentary strata in Germany. They combine this new data with previously-published information to extrapolate to the fluxes of K, Mg, and Fe into glauconite across Triassic through Holocene time and argue that authigenic glauconite formation has been a significant sink across this time period. In general, clays formed through reverse-weathering type reactions are (rightfully!) being increasingly recognized as important contributors to many elemental cycles in both the modern and deep time, so from that perspective I think this manuscript would generally be of broad interest to geochemists. However, I noticed some issues in the manuscript that merit revision prior to publication. Two of these I would consider major issues, and the rest are minor/moderate details; I describe these concerns below: We thank the reviewer for the overall positive evaluation of our work and acknowledge the improvement suggestions and comments we received, which clearly improved the manuscript. Accordingly, all issues raised by the reviewer have been implemented in the revised manuscript.

In general, the manuscript is heavy in self-citations (~20% of the citations were to previous work with Baldermann and/or Banerjee as lead or coauthor, but these citations are also the most heavily cited in the text). While I recognize that this degree of self-citation is often appropriate, particularly since both these authors have clearly conducted substantial previous work on the topic of glauconite formation, I questioned whether all these citations were necessary, and it was hard to tell how much of the work presented here was new, vs. reiterating conclusions already made in these previous publications. Self-citations have been reduced to a minimum. Several other papers (i.e., 16, from outside our group) have been cited that are related to (i) glauconite formation in marine sediments and (ii) global marine element cycles.

Furthermore, I think the heavy self-citation led to the authors incorrectly assuming some background and terminology would be familiar to readers (e.g. “evolved” and “highly evolved” glauconite), but it is really asking a lot of readers to expect that they go back and read all of the authors previous work to catch up on this terminology and context. If all these citations are truly needed, then the manuscript would benefit from the authors more clearly introducing terminology from previous works and more clearly differentiating which parts of the conclusions are novel. Glauconite terminology has been explained, i.e., “evolved” and “highly evolved” denote distinct stages of glauconite maturity, following the classification of Odin and Matter (1981). Self-citations have been reduced, in particular in the results. All conclusions drawn with respect to elemental fluxes related to glauconite formation through time are novel.

Finally, while the authors were liberal with self-citations, the manuscript was missing several key citations to important previous work on reverse weathering, in particular work by Rahman and colleagues using ^{32}Si to measure rates of reverse weathering, and the recent re-estimate of the marine silica cycle budget by Treguer et al. (2021). Not only were these citations missing, but the authors’ apparent unfamiliarity on important recent work on reverse weathering from the perspective of the silica cycle led them to make several factually incorrect statements (detailed below). I can mainly comment here on these missing or incorrect references with respect to Si because I am most familiar with it, but I wondered whether there might also be other key references missing in other parts of the paper. I’d suggest the authors should place

more emphasis on connecting their work to studies by researchers outside the co-author team. All references mentioned by the reviewer (as well as other relevant studies from outside our group) have been cited accordingly. We note here that we now also include Si (and Al) in our elemental flux calculations, as they are major elements in glauconite. The discussion and all relevant figures and SI Tables have been revised.

I also found the organization and logic of the discussion to be counterintuitive – the authors first extrapolate from a relatively small set of data points to make estimates of elemental sink sizes for the entire Mesozoic-Cenozoic time periods, then refocus on the modern system (which appears to be mainly recapitulating data from previous studies) to justify that these fluxes are important in the modern. It seems to me that a more logical approach would address the modern, which is theoretically better constrained first (although this is still extrapolated from only two modern sites that were presumably chosen because they have glauconite, so I question whether these are truly representative). Philosophically, the whole paper felt set up to inevitably lead to the conclusion that this process is important and didn't seem to be really set up to have the option to falsify the hypothesis that glauconite formation is significant. The structure of the discussion has been modified as requested. Based on the calculated elemental fluxes we indeed conclude that glauconite formation is important.

Specific comments:

Lines 54-61: this sentence is extremely long and was impossible to follow given the complex grammatical structure. Please split up into multiple shorter sentences. Done.

Lines 69-70: This could use some clarification - really excellent recent work by Rahman and others has made huge steps in better constraining reverse weathering fluxes (primarily on continental margins) which has essentially balanced the global marine Si budget (see Treguer et al 2021). So in that way, the magnitude is not "unknown". Are the authors referring more specifically to deep sea environments? We agree, and have cited these 2/3 papers accordingly. However, we note here that Rahman et al. (2016) concluded that “~0.4–0.5 Tmol/y of Si may be buried in these two deltaic dispersal systems alone, accounting for 30–50% of the global coastal Si sink presently attributed to authigenic clay burial in estuarine, shelf, and deltaic environments [Laruelle et al., 2009; Tréguer and De La Rocha, 2013] and implying that the latter is an underestimate”. Also, Rahman et al. (2017) state “Traditional bSi_{opal} and modified operational leaches designed to target the most reactive authigenic silicates ($\sim bSi_{altered}$) consistently underestimate authigenic clay formation (bSi_{clay}) and thus the magnitude of bSi_{total} burial in temperate coastal zones and subtropical deltas by 2–4-fold.” We do not cite Treguer et al. (2021) at this place, because this review paper (i) uses the Si fluxes reported in Rahman et al. (2016, 2017) for Si mass balance considerations and (ii) focuses more specifically on biogenic Si fluxes, paying only little attention to reverse weathering Si fluxes. Thus, we have modified the text as follows: “Moreover, the magnitude of element burial attributable to reverse weathering and (green) clay mineral authigenesis on the ocean floor remains poorly constrained, except for very specific settings (e.g., well studied deltaic sediments), and is only indirectly accounted for in earth system models (Arvidson et al., 2013; Rahman et al., 2016; Rahman et al., 2017; Isson and Planavsky, 2018).”

Line 74: here and in the discussion, please provide more context about these numbers. It's not entirely clear why a timescale unit is used to refer to something that has a rate. The timescales have been deleted to avoid confusion.

Lines 73-82: This paragraph is missing citations to the work of Rahman et al. using the ^{32}Si tracer to characterize these rates. Citations have been added.

Lines 96-97: Include the age of this specific section here in the text. The text has been changed as follows: “In this study, we fill this gap using a well-characterized, Cretaceous-aged glauconite-bearing sequence from Langenstein,...”.

Line 130 & Line 134: Please define the terms “evolved” and “highly evolved” glauconite as readers of this journal are unlikely to be familiar with them. The text has been changed as follows: “... which is characteristic of mature glauconite (i.e., reflecting the ‘evolved’ to ‘highly evolved’ stage, according to the glauconite maturity classification of Odin and Matter, 1981).”

Lines 211-216: Another area missing key references to other work while also citing multiple papers from the co-author team. Citations have been added (including Tréguer et al., 2021).

Line 225: this statement is factually incorrect about Si. Modern seawater productivity is closely tied to diatoms, who are using Si in seawater. It's not absent! Also, the most current global marine silica cycle model (Tréguer et al. 2021) indicates that reverse weathering is the 2nd largest sink for Si in seawater, so it's clearly important. Equating the relatively low concentration of Si in seawater to its relative unimportance in elemental cycling or with respect to reverse weathering is not correct and therefore not adequate justification to ignore Si in the analysis. We agree. This sentence has been deleted accordingly. In the second paragraph of the first discussion point we have provided the rationale for considering K and Mg (‘conservative elements’, primarily sourced from seawater), as well as Al, Si and Fe (‘scavenged elements’, mainly derived from marine sediment sources) in our revised elemental flux calculations.

Line 259: I’m confused by the reference to lower dissolved silica concentration in deep water settings because it is typically highest in deep waters as sinking diatom frustules dissolve. The reference to the lower silica concentration in the deep waters has been deleted.

Lines 265-268: The recent Tréguer et al 2021 Si budget should be cited here. Done.

Lines 297 and 299: why are “older” and “old” in quotations? Quotations have been deleted.

Figure 1: Text in panels A and C is illegibly small. Text size has been increased.

Reviewer #2 (Remarks to the Author):

This is a great manuscript. The manuscript is clear and was easy to review. I think that the data presented here supports their conclusions. Historically, reverse weathering has not been an easy process to track both in modern and paleo settings, and for that reason this work moves the field forward. Overall, I recommend publication with revisions detailed below. We thank the reviewer for the very positive evaluation of our work. All issues raised by the reviewer have been implemented in the revised manuscript.

General comment

Si release. Not a big deal, in Lines 151-152 the authors talk about Si liberation during substitution of Fe and Mg – associated with glauconite formation. This is potentially important for Si cycling and the global Si budget. Would it be possible to estimate the flux of Si liberated through glauconite formation in this context? The substitution reactions mentioned here do not imply Si removal from the glauconite crystal lattice during glauconitization. Indeed, silicic acid is consumed during glauconite formation, according to the expression: Fe-smectite + silicic acid + goethite (or a similar phase) + K^+ + Mg^{2+} + Fe^{2+} -> glauconite + Na^+ + Ca^{2+} + H_3O^+ (see Eq. 6 in Baldermann et al., 2017, or Fig. 10 in Fernández-Landero and Fernández-Caliani, 2021). For this reason, and following the suggestion of reviewer 1, we have calculated the Si and also Al fluxes related to glauconite formation through time. The rationale for considering K and Mg ('conservative elements', primarily sourced from seawater) and Al, Si and Fe ('scavenged elements', mainly derived from marine sediment sources) is provided in the second paragraph of the first discussion point. The discussion and all relevant Figures have been updated.

Sedimentation rate. A single sedimentation rate of 130 cm·Myr⁻¹ was used in this study. This is a critical parameter to determine the sequestration rate of the different elements. Is there a range (upper and lower limit) that could be used here to better represent possible sequestration rates of K, Mg and Fe? We fully agree that the sedimentation rate is a critical parameter. We have now considered a range of sedimentation rates from 0.1 to 100 cm/kyr for the palaeo-flux calculations (see Fig. 5), consistent with published literature for shelfal settings (Piñero et al., 2013; Dutkiewicz et al., 2017). As for the Langenstein section, we have chosen a low sedimentation rate for the glauconite sandstone (0.13 cm/kyr, representing the lower part of the M. dixonii zone), and a moderate sedimentation rate for the glauconitized limestones (7 cm/kyr, representing the middle and upper part of the M. dixonii zone), consistent with Wilmsen (2003, 2007). For the present-day shelf regions, we also assume a moderate sedimentation rate of 10-20 cm/kyr vs 0.4-0.8 cm/kyr, reflecting glauconite formation in many modern shelfal vs deep-sea sediments. However, Restrepo et al., (2020), which we also cite in the paper, indicates that present-day shelf areas can display sedimentation rates between 0.1 and 1.0 cm/yr; these seem to be too high to be representative of glauconite formation in modern shelfal sediments (Odin and Matter, 1981; Chatteraj et al., 2016; Fernández-Landero and Fernández-Caliani, 2021). The discussion and all relevant Figures have been updated.

Low versus high sedimentation (Line 186 and 240). I am curious, does low sedimentation rate really 'favor' glauconite formation (i.e., elevated rate of formation) or is this simply a dilution effect, with higher sedimentation rates giving rise to higher dilution and therefore lower abundances of glauconite expressed. The residence time at the sediment-seawater interface mainly controls the abundance of glauconite in marine sediments, which is often predefined by the detrital sediment input flux vs. marine carbonate bio-productivity (i.e., the sedimentation rate). We note here that glauconitization describes a process that preferentially occurs within a semi-confined micromilieu (foraminifera chambers, fecal pellets, etc.) soon after sediment deposition: Charpentier et al. have calculated that a few kyr are needed to produce Fe-smectite, which subsequently alters into glauconite via the formation of mixed-layer glauconite-smectite over a few Myr. Continuous supply of all major elements (K, Mg, Al, Si and Fe) to the active

glaucanization front is required to run the reaction to completion, which is usually the case when glauconite growth takes place immediately at the sediment-seawater interface at low to moderate (reduced) sediment accumulation rates. Therefore, low sedimentation rates help to increase the abundance and maturity of glauconite minerals in marine sediments rather than the glauconite formation rate.

Line 60: other relevant citations here could include (Isson et al., 2020; Kump et al., 2000). We have cited Isson et al. here, because it is the most recent work.

Line 68: first mention of potassium (K), might be best to spell this out? Done.

Line 82: modern... (relevant citations (Ehlert et al., 2016)) and ...ancient marine sediments (relevant citations here (Kalderon-Asael et al., 2021; Li et al., 2021)). These papers have been cited accordingly.

Line 208: why not just recycling? Changed to recycling.

Line 254: ... shelfal sequestration rates. Requires citation here. Shelfal sequestration rates have not been published before. We have changed the text as follows: "(...) ~50-130-times lower than the Langenstein sequestration rate."

References We thank the reviewer for providing these references.

Ehlert, C., Doering, K., Wallmann, K., Scholz, F., Sommer, S., Grasse, P., Geilert, S., & Frank, M. (2016). Stable silicon isotope signatures of marine pore waters—Biogenic opal dissolution versus authigenic clay mineral formation. *Geochimica et Cosmochimica Acta*, 191, 102-117.

Isson, T., Planavsky, N., Coogan, L., Stewart, E., Ague, J., Bolton, E., Zhang, S., McKenzie, N., & Kump, L. (2020). Evolution of the Global Carbon Cycle and Climate Regulation on Earth. *Global Biogeochemical Cycles*.

Kalderon-Asael, B., Katchinoff, J. A., Planavsky, N. J., Hood, A. v. S., Dellinger, M., Bellefroid, E. J., Jones, D. S., Hofmann, A., Ossa, F. O., Macdonald, F. A., Wang, C., Isson, T. T., Murphy, J., Higgins, J., West, J., Wallace, M., Asael, D., & Pogge von Strandmann, P. (2021). A lithium-isotope perspective on the evolution of carbon and silicon cycles. *Nature*, 595(7867), 394-398.

Kump, L. R., Brantley, S. L., & Arthur, M. A. (2000). Chemical weathering, atmospheric CO₂, and climate. *Annual Review of Earth and Planetary Sciences*, 28(1), 611-667.

Li, F., Penman, D., Planavsky, N., Knudsen, A., Zhao, M., Wang, X., Isson, T., Huang, K., Wei, G., & Zhang, S. (2021). Reverse weathering may amplify post-Snowball atmospheric carbon dioxide levels. *Precambrian Research*, 364, 106279.

REVIEWERS' COMMENTS

Reviewer #1 (Remarks to the Author):

I commend the authors on thoroughly addressing most of my concerns; the revised paper is a clear improvement on what was already strong work. I only have a few minor comments for clarity that I would suggest the authors address, but these constitute a very minor revision.

Lines 69-70: Is there a noun missing here after "hydrothermal"?

Lines 115-116, 119-120: I appreciate that the authors have expanded a bit on the meaning of some of these terms and I think I generally follow the meaning of "maturity" in this context (especially from referring to the figure), but I think given the broad readership of Nat Comms, it would still be useful to define what "maturity" means in this context.

Lines 295-296: In this section and in the Fe section below, I'm not sure I follow how the authors are mapping their estimates to the atmospheric and glacial fluxes. Are these just percentages of the deep sea flux? And if so, I don't know that it's strictly correct that all the glacial Si fluxes end up in deep seawater? I think these sections could use a little clarity for which fluxes the authors are comparing to their shelfal vs deep sea flux estimates.

Fig 3 - the placement and size of the A-E letter labels in the figure threw me off - they look awkwardly large relative to the size of the font in the rest of the figure and since some labels are not closed to the part of the figure they refer to (E) or not underneath the section they refer to (A), I found it visually confusing. I'm not sure a figure like this actually needs letters like this since the caption could be slightly re-written to just state that datasets are described from left-to-right. I'd suggest either making that change and omitting labels or rethinking the design a little to make the figure more polished and easier to follow.

Reviewer #1 (Remarks to the Author):

Lines 69-70: Is there a noun missing here after "hydrothermal"? We have added 'settings' after hydrothermal.

Lines 115-116, 119-120: I appreciate that the authors have expanded a bit on the meaning of some of these terms and I think I generally follow the meaning of "maturity" in this context (especially from referring to the figure), but I think given the broad readership of Nat Comms, it would still be useful to define what "maturity" means in this context. We have provided the following explanation at the start of the paragraph: "Glauconite commonly evolves from an authigenic Fe-smectite precursor⁴⁶, with glauconite maturation from a K-poor but Fe(III) rich nascent stage (< 4 wt.% K₂O) to a K-rich evolved or highly evolved stage (> 8 wt.% K₂O) occurring over less than a few Myr."

Lines 295-296: In this section and in the Fe section below, I'm not sure I follow how the authors are mapping their estimates to the atmospheric and glacial fluxes. Are these just percentages of the deep sea flux? And if so, I don't know that it's strictly correct that all the glacial Si fluxes end up in deep seawater? I think these sections could use a little clarity for which fluxes the authors are comparing to their shelfal vs deep sea flux estimates. In this section we do not separately consider deep sea and shallow water fluxes, mainly because the deep-sea flux is comparatively negligible. However, to avoid confusion, we have replaced "as well as" by "or". We have also slightly modified the second last sentence in the Fe paragraph for clarity.

Fig 3 - the placement and size of the A-E letter labels in the figure threw me off - they look awkwardly large relative to the size of the font in the rest of the figure and since some labels are not closed to the part of the figure they refer to (E) or not underneath the section they refer to (A), I found it visually confusing. I'm not sure a figure like this actually needs letters like this since the caption could be slightly re-written to just state that datasets are described from left-to-right. I'd suggest either making that change and omitting labels or rethinking the design a little to make the figure more polished and easier to follow. The figure has been revised accordingly (i.e., the size and the positioning of the labels were changed).